# Pattern of Antibiotic Use among Hospitalized Patients according to WHO Access, Watch, Reserve (AWaRe) Classification: Findings from a Point Prevalence Survey in Bangladesh

**DOI:** 10.3390/antibiotics11060810

**Published:** 2022-06-16

**Authors:** Md. Mahbubur Rashid, Zubair Akhtar, Sukanta Chowdhury, Md. Ariful Islam, Shahana Parveen, Probir Kumar Ghosh, Aninda Rahman, Zobaidul Haque Khan, Khaleda Islam, Nitish Debnath, Mahmudur Rahman, Fahmida Chowdhury

**Affiliations:** 1Programme for Emerging Infections, Infectious Diseases Division, International Centre for Diarrhoeal Disease Research, Bangladesh (icddr,b), Dhaka 1212, Bangladesh; zakhtar@icddrb.org (Z.A.); sukanta@icddrb.org (S.C.); arif@icddrb.org (M.A.I.); shahana@icddrb.org (S.P.); probir@icddrb.org (P.K.G.); fahmida_chow@icddrb.org (F.C.); 2School of Women’s & Children’s Health, Faculty of Medicine and Health, University of New South Wales (UNSW), Sydney, NSW 2052, Australia; 3Biosecurity Research Program, The Kirby Institute, University of New South Wales (UNSW), Sydney, NSW 2052, Australia; 4Communicable Disease Control (CDC), Directorate General of Health Services, Government of Bangladesh, Dhaka 1212, Bangladesh; dr.turjossmc@gmail.com; 5Fleming Fund Country Grant to Bangladesh, DAI Global, LLC, House 3, Road 23B, Gulshan 1, Dhaka 1212, Bangladesh; zobaid_khan@dai.com (Z.H.K.); dr.khaleda.islam@gmail.com (K.I.); nitish_debnath@dai.com (N.D.); 6Global Health Development, EMPHNET, 69 Mohakhali, Dhaka 1212, Bangladesh; mahmudur57@gmail.com

**Keywords:** AWaRe, WHO, antibiotics, access, watch, Bangladesh, hospitals

## Abstract

For supporting antibiotic stewardship interventions, the World Health Organization (WHO) classified antibiotics through the AWaRe (Access, Watch, and Reserve) classification. Inappropriate use of antimicrobials among hospital-admitted patients exposes them to the vulnerability of developing resistant organisms which are difficult to treat. We aimed to describe the proportion of antibiotic use based on the WHO AWaRe classification in tertiary and secondary level hospitals in Bangladesh. A point prevalence survey (PPS) was conducted adapting the WHO PPS design in inpatients departments in 2021. Among the 1417 enrolled patients, 52% were female and 63% were from the 15–64 years age group. Nearly 78% of patients received at least one antibiotic during the survey period. Third-generation cephalosporins (44.6%), penicillins (12.3%), imidazoles (11.8%), aminoglycosides (7.2%), and macrolides (5.8%) were documented as highly used antibiotics. Overall, 64.0% of Watch, 35.6% of Access, and 0.1% of Reserve group antibiotics were used for treatment. The use of Watch group antibiotics was high in medicine wards (78.7%) and overall high use of Watch antibiotics was observed at secondary hospitals (71.5%) compared to tertiary hospitals (60.2%) (*p*-value of 0.000). Our PPS findings underscore the need for an urgent nationwide antibiotic stewardship program for physicians including the development and implementation of local guidelines and in-service training on antibiotic use.

## 1. Introduction

Irrational and inappropriate use of antimicrobials specifically antibiotics in humans, animals, and in the food chain leads to the acceleration of the emergence of antimicrobial resistance (AMR) or antibiotic resistance (ABR). A large proportion of the burden of infections occurs by resistant pathogens due to healthcare-associated infections (HAIs) [1]. The increasing trend of resistant organisms subsequently leads to treatment failure, significant morbidity and mortality, and poses additional out-of-pocket expenditure and healthcare costs annually [2,3]. In 2019, about 4.95 million deaths were estimated to be associated with bacterial AMR and 1.27 million deaths were attributable to bacterial AMR [4]. Currently, the use of antibiotic consumption is escalating worldwide driven by rising purchasing capacity, health insurance, and the burden of various infectious diseases [5]. The overuse of antimicrobials in outpatients was also documented in Bangladesh due to both self-medication and physician’s prescriptions [6,7,8]. The problem necessitates two ways to draw attention: interrupting the transmission of resistant organisms and cautious use of antimicrobials in hospitals [9].

In the 68th World Health Assembly, the highest importance was given to AMR considering the health and economic consequences of adopting the global action plan for all member states [10]. The World Health Organization’s (WHO) expert committee developed the ‘AWaRe’ classification on using the Essential Medicine List (EML) in 2017 to support, strengthen, and monitor the antibiotic stewardship program and was further updated in 2021 [11]. The goal is to ensure the total consumption of Access group antibiotics at least 60% at the country level, and the index also helps to calculate the comparative use of narrow-spectrum and broad-spectrum antibiotics [12,13]. To understand the global prevalence of antimicrobial use and resistance with special emphasis on Low- and Middle-Income Countries (LMIC), the Global Point Prevalence Survey (GLOBAL-PPS) of antimicrobial use and resistance was developed after the 4th World Healthcare-Associated Infections and Antimicrobial Resistance Forum [14,15]. In accordance with the WHO, Bangladesh also adopted the Global Action Plan (GAP) and developed a National Action Plan (NAP) 2017–2022 for the containment of AMR. Although the country has started a surveillance system to generate reliable data on AMR, there remains a dearth of information on antimicrobial usage (AMU) or antibiotic usage (ABU). There is also a paucity of information regarding the antimicrobial prescription at inpatient departments which is a daunting barrier to the successful development and implementation of antimicrobial stewardship programs. Aligning with the Global-PPS and the WHO, a point prevalence survey was conducted at both government tertiary and secondary level acute care hospitals in Bangladesh. The objectives of the survey were to understand the prevalence of AMU, the distribution of antibiotic agents used, to improve AMU knowledge, and to identify the possible scope of intervention for promoting prudent use of antibiotics utilizing WHO’s ‘AWaRe’ classification.

## 2. Methods

### 2.1. Study Design and Study Population

The point prevalence survey on AMU among inpatients was the part of a larger project having the objective of understanding antimicrobial use in humans (at hospitals, communities, and pharmacies), commercial chicken, and aquaculture in Bangladesh through the ‘One Health’ approach. The survey was conducted in four acute care government hospitals (two tertiary and two secondary level hospitals) selected purposively in 2 administrative divisions of Bangladesh from 18 February to 6 April 2021. A structured survey questionnaire was adapted from WHO and Global-PPS design to assess the extent of antimicrobial prescribing patterns among the hospitalized patients at different wards of survey hospitals [16,17]. The survey was conducted among inpatients of medicine, surgery, gynae and obstetrics, and pediatric wards including intensive care units.

### 2.2. Inclusion Criteria

All patients admitted in the ward on the day of the survey from Saturday to Thursday in a week before 08:00 o’clock were included in the survey upon obtaining written informed consent from patients or caregivers [16,17].

### 2.3. Exclusion Criteria

Patients from outpatients or day-care patient facilities, nursing homes, psychiatric wards, long-term care wards, emergency departments, out-patient dialysis, discharged patients waiting for parents or relatives, and outpatient parenteral antimicrobial therapy were excluded from the survey [16,17]. Moreover, patients admitted in the selected wards after 8.00 o’clock or transferred from other wards to the selected wards after 8.00 o’clock, patients’ receiving topical and ophthalmologic antimicrobials, and treatment initiated after 08:00 o’clock on the day of survey were excluded.

### 2.4. Data Collection and Study Variables

Study physicians collected patient’s demographic and clinical information, indications for antimicrobial use, microbiological lab findings, and details of antimicrobial used. For secondary level hospitals (≤500 bedded), all eligible patients were enrolled, and for tertiary level hospitals (≥800 bedded), every 3rd patient of each ward was enrolled after securing a written informed consent. If a patient refused to join the survey, the next patient on the list was approached. A single hospital was completed within a maximum of three consecutive weeks from the onset of the first day of data collection with a special emphasis on one ward in one day to minimize the impact of patients moving between wards. For antimicrobial data, the study focused on antibacterials for systemic use, antiparasitics and antifungals for systemic use, antiprotozoals used as antibacterial agents, imidazoles derivatives, all antivirals, and antimalarials.

The indications for antimicrobial use were set in four categories such as Community Acquired Infection (CAI), Hospital Acquired Infection (HAI), Medical Prophylaxis (MP), and Surgical Prophylaxis (SP). CAI was considered if a patient was admitted in a hospital from home or their community with infection or symptoms of infection [5]. Moreover, if a patient developed an infection or symptoms of infection after >48 h of admission, they were registered as HAI [18]. Medical Prophylaxis (MP) was defined as antimicrobials which were used to prevent an infection in inpatients with medical conditions in absence of any diagnosis of communicable disease [5]. Similarly, SP was identified if antimicrobials were used to prevent infection at surgical sites [19]. The study physicians explored indications for antimicrobial use by interviewing patients or caregivers or treating doctors and by assessing the patients’ treatment sheets. Moreover, SP was classified in three categories depending on doses and duration on the day of survey; one dose of antimicrobial used for one day/multiple days (SP1), multiple doses used for one day (SP2), and multiple doses for more than one day (SP3). Patients’ age categories were adapted form Management Information System of the Directorate General of Health Services (DGHS) [20] where patients aged below 14 years are generally admitted in pediatric departments in government hospitals of Bangladesh.

### 2.5. Data Management and Statistical Analysis

The data were analyzed using the statistical software STATA 13.0 (Special edition). For descriptive analysis, hospital information, ward data, demographic and clinical information of patients, antimicrobial data were analyzed and illustrated in figures and tables. Continuous variables were reported by means (standard deviation) or medians (interquartile range). Bi-variate analysis was done using chi-square test to understand association among antibiotics use, sex, age-group, departments, history of surgery, use of devices, history of transfer from another hospital, and indications of antimicrobial use. Based on the findings of bivariate analysis and significance level *p* ≤ 0.25, we performed multivariable logistic model to identify the independent associated variables. The results were reported as adjusted odds ratio (aOR) with a 95% confidence interval (CI). The distribution of antibiotics was analyzed using WHO AWaRe classification, 2021 [11].

### 2.6. Ethical Consideration

The Institutional Review Board (IRB) of International Centre for Diarrhoeal Disease Research, Bangladesh (icddr,b), granted the approval to conduct the study after protocol review by the research review committee and ethical review committee. Permission and necessary support to implement this study at government hospitals were received from DGHS of Bangladesh Government. Informed written consents from patients or caregivers of pediatric and intensive care unit patients were obtained before commencing the data collection by explaining the purpose and scope of the study, assurance of the confidentiality of personal health information, and clarifying any questions.

## 3. Results

A total of 3140 eligible patients were admitted during the study period; among them, 892 patients (34.3%) were enrolled from 2599 eligible patients in two tertiary level hospitals and 525 patients (97.0%) were enrolled from 541 eligible patients in two secondary level hospitals. Thereby, the enrolment rate was 45.1% (1417/3140).

### 3.1. Demographic and Clinical Characteristics

Among the surveyed patients, female patients constituted 51.6%; 62.6% of patients were from the 15–64 years age group and 28.2% were from the 0–14 years age group which belong to the pediatric population (Table 1). Mean age of the enrolled patients was 31.1 years (SD: ±23.3). Overall, 49% of patients had primary to higher level of formal education. The majority of patients (36.1%) were involved in household work, 13.6% were farmers, and 28.4% were unemployed. Among the enrolled patients, 33.4% were from medicine wards, 28.2% from surgery wards, 22.4% from pediatric wards, and 15.9% from gynae and obstetric wards.

Nearly one-sixth of patients were transferred from other hospitals and one-seventh of patients had hospitalization history in the last three months prior to the current hospital admission (Table 1). A peripheral vascular catheter was used among 73.9% of patients and 9.2% of patients had a urinary catheter (Table 1). According to the indication of antimicrobial use, 47.7% of patients received MP; 31.8% of patients received antimicrobials for CAI. SP was found among 26.7% at tertiary and 16.6% at secondary level hospitals (Table 1). Among the patients with SP, 92.8% received multiple antimicrobials for more than one day.

### 3.2. Antibiotics Usage According to Demographic and Clinical Characteristics

Among the surveyed patients, 74.3% and 82.3% of female patients at tertiary and secondary level hospitals received at least one antibiotic, respectively (Table 2); and 90% of patients from the 0–14 years age group, 73.8% of the 15–64 years age group, and 64.9% of the over 64 years age group received at least one antibiotic. Proportioning for different departments, 91.2% of pediatric patients, 81.4% of gynae and obstetric patients, 78.0% of surgery patients, and 66.2% of medicine patients received at least one antibiotic. According to our findings, 98.2% of patients who had surgery after admission received antibiotics. For devices used, 86.9% of patients with urinary catheter, 84.5% of patients with peripheral vascular catheter, 80% of patients with intubation device, and 73.3% of patients with central vascular catheter received antibiotics (Table 2). Moreover, 83.7% of patients who were referred from other hospitals and 71.6% of patients with previous hospitalization in the last three months had at least one antibiotic. Overall, 81.7% of patients received at least one antimicrobial and 77.6% of patients received at least one antibiotic. On average, every patient received 1.2 (SD: ±0.53) antibiotics during the survey period.

### 3.3. Antimicrobials Used

According to the PPS, 98.8% of antimicrobials were antibiotics used during survey period and rest were antivirals (0.9%), antiparasitics (0.2%), and antifungals (0.1%). A total of 2138 encounters of antimicrobials were recorded among 1157 patients through the survey both in tertiary and secondary level hospitals (Table 3).

### 3.4. Cephalosporin Groups

Among the antibiotics, the cephalosporin group constituted 51.4% (1086/2112) and the use of it in all departments was 48.9% (693) and 54.4% (393) in tertiary and secondary level hospitals, respectively (Table 3). The uses of third generation cephalosporins were documented as 49.8% (304), 46.9% (217), 47% (275), and 33% (158), respectively, in surgery, medicine, pediatrics, and gynae and obstetric wards. Among the cephalosporin groups, second generation cephalosporins (4.3%; 93) was the second highest used cephalosporin after the third generation. Moreover, the use of fourth generation cephalosporins was higher in pediatric wards (1.2%) compared to other wards (Table 3).

### 3.5. Other Groups of Antibiotics in Different Departments

Other than cephalosporins, penicillins (12.3%; 263), imidazoles (11.8%; 253), aminoglycosides (7.2%; 154), macrolides (5.8%; 124), fluoroquinolones (4.5%; 96), and carbapenems (3%; 64) were recorded as top few antibiotics used in the tertiary and secondary level hospitals. The use of macrolides was 17.7% (82) in medicine wards. In pediatric wards, use of aminoglycosides and carbapenems were documented as 22.6% (132) and 7% (41), respectively (Table 3).

### 3.6. Number of Antibiotics Used for Treating Patients after Admission

Among 1417 enrolled patients, 62.2% (882) of patients received one antibiotic that is one antibacterial agent, 12.7% (180) received two antibiotics, 2.2% three antibiotics (31), 0.4% (5) four antibiotics, and 0.1% (1) five antibiotics from admission to the date of survey (Figure 1). Use of two or more antibiotics was found high in gynae and obstetric (28.3%; 64), surgery (15.7%; 63), and pediatric wards (13.8%; 44).

### 3.7. Antibiotics Used According to AWaRE Category

According to the WHO AWaRe category, Watch group of antibiotics were used in 64.0% (1352) of patients followed by 35.6% (752) of Access and 0.1% (2) of Reserve group of antibiotics in the selected hospitals (Figure 2). In medicine wards, 78.7% (358) of patients were treated with Watch antibiotics and 20.7% (94) with Access antibiotics (Figure 2). About 62.9% (382), 36.4% (221), and 0.2% (1) of patients were treated with Watch, Access, and Reserve group of antibiotics, respectively, in surgery wards. Among the patients of gynae and obstetric wards, 55.4% (265), 44.4% (212), and 0.2% (1) patients were treated with Watch, Access, and Reserve antibiotics, respectively. Moreover, among pediatric patients, 60.7% (347) and 39.3% (225) of children received Watch and Access antibiotics, respectively. Use of Watch antibiotics was found more in secondary hospitals (71.5%) compared to tertiary hospitals (60.2%) (*p*-value of 0.000). On the contrary, Access antibiotics were used more in tertiary hospitals (39.4%) compared to secondary hospitals (28.3%).

### 3.8. Assiciation between Antibiotics Use (According Aware Classification) and Characteristics of Patients and Hospitals

In the multivariable logistics model, we found that the Watch antibiotics were more likely to be used among children aged under 14 years (aOR = 6.96, 95% CI: 3.22, 15.04), patients having surgery since admission (aOR = 19.43, 95% CI: 6.99, 54.04), and patients used peripheral vascular catheter (aOR = 4.85, 95% CI: 3.52, 6.69), while it was less likely to be used in the medicine wards (aOR = 0.47, 95% CI: 0.29, 0.77) (Figure 3B). The Access antibiotics were more likely to be used among patients who had surgery since admission (aOR = 12.91, 95% CI: 8.07, 20.65), use of peripheral vascular catheter (aOR = 3.09, 95% CI: 2.24, 4.26), and patients transferred from another hospitals (aOR = 1.88, 95% CI: 1.36, 2.57) (Figure 3A).

## 4. Discussion

Our survey substantiated that the lion’s share of the antibiotics was used from the Watch group of the WHO Essential Medicine List (EML) based on empirical treatment. The use of Watch group antibiotics was observed in 64.0% among overall patients with higher use in secondary hospitals (71.5%) compared to tertiary hospitals (60.2%). Based on the target set by WHO for using Access group of antibiotics for 60% of patients, a reverse usage of Watch antibiotics was observed in the survey. Although, all the study hospitals had laboratory capacities, the test-based directed antimicrobial use was limited. Time consumption for the culture of bio-samples might hinder the directed treatment in hospitals. The high burden of antibiotic use also indicates a lack of implementation of guidelines in the study hospitals as most of the hospitals have no internal guidelines, and existing national guidelines are focused on few specific diseases. The study also observed the use of Reserve category antibiotics in surgery and gynae and obstetric wards which was lower than India, high-income, upper-middle-income-, and lower-middle-income countries [21,22]. The lower use of Reserve group of antibiotics in Bangladesh is an opportunity to restrict the use by reducing commercial availability and prescription practices. The PPS was conducted following WHO PPS methodology and to the best of our knowledge, this was the foremost analysis of antibiotics in the inpatient departments of government health facilities in Bangladesh based on WHO ‘AWaRe’ classification. The prevalence of antibiotic use documented by this study was commensurable to the findings of Nigeria (81%) and Pakistan (77.6) which was higher than Botswana (70.6%), India (50%), and Ghana (51%) [21,23,24,25,26]. The findings of the survey were much higher than Southern Europe (39%), North America (37%), and the global prevalence (34%) reported by the study of Versporten et al. [14].

It is noteworthy to state that a remarkable proportion of patients received two or more antibiotics during their hospital stay at the time of survey which was lower than Ghana and Botswana [23,25]. Use of multiple antibiotics and use of two to five antibiotics in a single admission should be an important concern. The high burden of antibiotic use was observed in pediatric wards specifically in secondary level or district hospitals compared to tertiary level hospitals. In an infant hospital in Bangladesh, the prevalence of antibiotic use was 73% [27]. This may be hypothesized accounting the overuse of antibiotics among pediatric patients. Firstly, physicians might think that pediatric groups are more susceptible to infection and hospitals are the reservoir of medically concerned infections. Secondly, due to the lack of proper understanding about antimicrobials and antibiotics, parents might request or demand for newer or expensive antibiotics for their children [28].

In our PPS, the proportion of CAI was 31.8% which was similar to Pakistan (34.2%) but lower than the Global PPS with an overall prevalence of 46% [14,26]. This might be due to healthcare delivery for common CAIs (pneumonia and diarrhea) are carried out at primary level hospitals and by general practitioners (GP) in Bangladesh. In addition, about 71% of encounters of GPs were prescribed with antimicrobials [29]. Moreover, due to the pandemic of COVID-19, patient-flow might be reduced considerably. HAI was considered as one of the crucial factors for antimicrobial use though the study documented very limited frequency. This might happen due to the nationwide COVID-19 pandemic, as the general population and hospital environment were forced to adopt some infection prevention and control measures including prophylactic antibiotic use [30,31]. Due to ‘No Mask, No Service’ initiative of the government of Bangladesh during COVID-19 pandemic, the majority of the patients used mask and practiced hand washing utilizing hand sanitizer or soap [32]. Patients’ admission was also reduced compared to earlier years. In addition to this, due to the fear of COVID-19, there might be an increased use of antibiotics as medical prophylaxis which might reduce the HAI considerably [31]. The similar prevalence of antimicrobial use as medical prophylaxis was found in Pakistan (57.4%) [26], hence, there are research gaps regarding appropriate indications for antimicrobial use as medical prophylaxis [33]. Regarding SP, a single dose of antibiotic for 24 h is sufficient for preventing infections [34,35]. Nevertheless, multiple doses of antibiotics as SP for more than one day was found significantly high and the similar result was found in a Saudi Arabia (85%) [36]. The higher use of SP revealed the likelihood of practicing over-prescription which increased the financial burden to hospitals, out of pocket expenditure of patients, and antibiotic resistance to clinically concerned microorganisms. Proper administration of antibiotic prophylaxis in surgery is very much important in hospitals and requires constant efforts and collaboration between prescribers and antimicrobial stewardship program [37].

Among the Watch group antibiotics, third generation cephalosporins, ceftriaxone was remarkably high among the antibiotics used in surgery wards. The findings were higher compared with a recent study of India (24.5%) and reports of worldwide use (24.2%) published in 2021 [21,22]. Besides, the excessive use of third generation cephalosporins (Watch) is alarming as it is considered as one of the factors of extended spectrum beta-lactamase (ESBL) producing microorganisms [38,39].

Among the top five antibiotics found in the PPS of Bangladesh, three were from the Watch group such as third generation cephalosporins, macrolides, and fluoroquinolones. Among them, macrolides could not score the position in the top five in many countries including India [14,21,22]. Macrolides are used commonly for gastrointestinal diseases and respiratory infections in Bangladesh, though the excessive use of macrolides may lower the susceptibility of *Streptococcus pneumoniae* and typhoidal *Salmonella* strains [22,40,41,42].

The ‘AwaRe’ classification of WHO develops a general guideline on antibiotic prescribing pattern in the health facilities and the objective is to improve the monitoring system on antibiotic use. The tool needs to be adopted by countries and hospitals for rational use of antibiotics and to fight the Global combat against AMR as the tool identifies antibiotics for empiric treatment or reserving as the last hope [43,44]. However, Budd et al. showed that the use of ‘AwaRe’ Classification in England improved the use of Access antibiotics [45]. As a global antibiotic stewardship tool, WHO AWaRe classification of EML should be adopted in all levels of health facilities. All the clinicians of tertiary level and secondary level hospitals should be informed and trained on current scenarios of ABU, irrational use of antibiotics, importance of AWaRe classification to limit irrational antibiotic use in government health facilities. Directorate General of Drug Administration (DGDA) and pharmaceutical companies may have a major role in controlling irrational antibiotic use. Laboratory facilities of the hospitals should be vitalized and properly equipped to support the inpatient departments for directed evidence-based ABU.

The survey had some limitations. The findings from the two tertiary and two secondary level hospitals cannot be generalized for the country or this region. For the representativeness at the country level, the survey should be conducted systematically in a population of representative sample all over Bangladesh and the results should be shared with special attention to hospital authorities, pharmaceutical companies, and policy makers. Due to the design of the study, the survey did not capture the seasonality of the antibiotic prescribing patterns. Finally, factors related with institution-wise antimicrobial resistance data, burden of infectious diseases, and supply of antibiotics were not considered in the PPS which might provide a different but valuable interpretation.

## 5. Conclusions and Recommendations

The PPS facilitated a benchmark on antibiotic usage at tertiary and district level hospitals illustrating excessive prescription of wide range of ‘Watch’ group of antibiotics among all age groups in Bangladesh indicating an irrational use which may facilitate extensive antibiotic resistance. For a better understanding of AMU, a robust PPS is highly recommended involving all administrative divisions of Bangladesh. The findings from this PPS at hospitals underscored the urgent need of a national antimicrobial stewardship program for promoting rational and directed antibiotic practice along with the development of local guidelines based on resistance patterns with a special emphasis on WHO AWaRe Classification, awareness building, frequent in-service training on AMR,, and proper law enforcement. Moreover, the Point prevalence survey should be repeated yearly to understand the progress and monitor AMU in Bangladesh considering this study as a baseline.

## Figures and Tables

**Figure 1 antibiotics-11-00810-f001:**
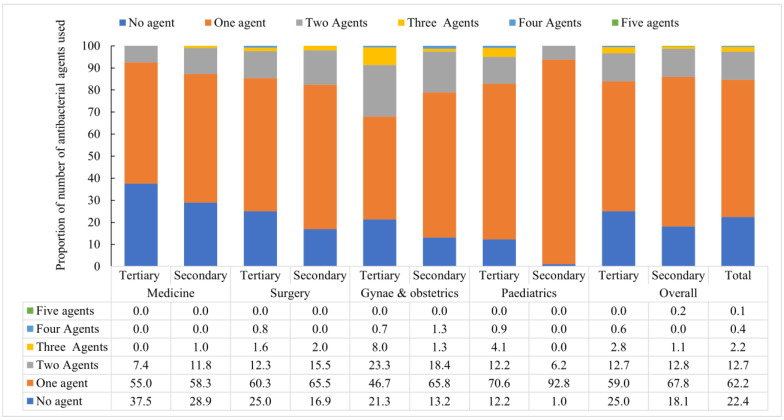
Number of antibacterial agents used for treating patients during hospitalization on the day of the point prevalence survey at different departments of tertiary and secondary level hospitals in Bangladesh from February to April 2021.

**Figure 2 antibiotics-11-00810-f002:**
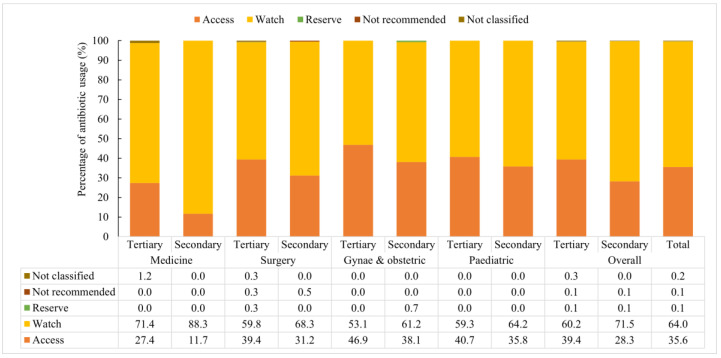
Antibiotics usage according to the WHO AWaRe classification at different departments of tertiary and secondary level hospitals in Bangladesh from February to April 2021.

**Figure 3 antibiotics-11-00810-f003:**
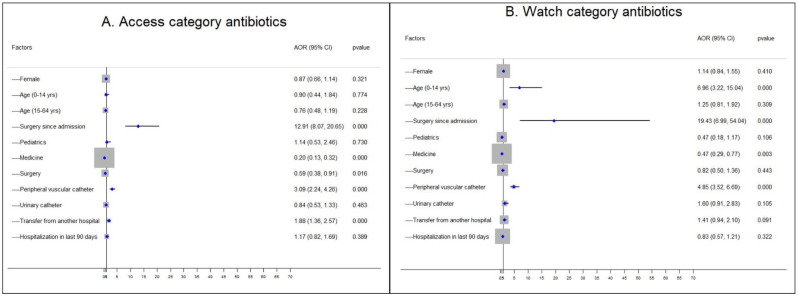
(**A**,**B**): Multivariate logistic regression model for the association between Access and Watch-group antibiotics and baseline covariates at tertiary and secondary level hospitals in Bangladesh from February to April 2021.

**Table 1 antibiotics-11-00810-t001:** Demographic and clinical characteristics of the patients enrolled in the point prevalence survey of antimicrobial usage at tertiary and secondary level hospitals in Bangladesh from February to April 2021.

Characteristics	Tertiary Level Hospitals (N = 892)	Secondary Level Hospitals (N = 525)	Overall (N = 1417)
	*n* (%)	*n* (%)	*n* (%)
**Sex**			
Female	455 (51.0)	276 (52.6)	731 (51.6)
Male	437 (48.9)	249 (47.43)	686 (48.4)
**Age group**			
0–14 years age group	251 (28.1)	148 (28.2)	399 (28.2)
15–64 years age group	561 (62.9)	326 (62.1)	887 (62.6)
65 years and above	80 (9.0)	51 (9.7)	131 (9.2)
**Mean age ± SD**	31.2 (23.1)	30.9 (23.5)	31.1 (23.3)
**Education level**			
No education	453 (50.8)	270 (51.4)	723 (51.0)
Primary	181 (20.3)	80 (15.2)	261 (18.4)
Secondary	176 (19.7)	116 (22.1)	292 (20.6)
Higher	82 (9.2)	59 (11.2)	141 (10.0)
Patient transfer from another hospital	218 (24.4)	39 (7.4)	257 (18.1)
Patient hospitalized in last 3 months	138 (15.5)	63 (12.0)	201 (14.2)
**Invasive device used**			
Central vascular catheter	11 (1.2)	4 (1.0)	15 (1.1)
Peripheral vascular catheter	649 (73.0)	398 (75.8)	1047 (73.9)
Urinary catheter	103 (11.6)	27 (5.1)	130 (9.2)
Intubation tube	4 (0.5)	1 (0.2)	5 (0.4)
Had surgery after admission	165 (18.5)	57 (10.9)	222 (15.7)
Patients on antimicrobials	716 (80.3)	441 (84.0)	1157 (81.7)
Patients on antibiotics	669 (75.0)	430 (81.9)	1099 (77.6)
Average number of antimicrobials used among patients since admission			
Mean (±SD)	1.9 (±1.06)	1.6 (±0.80)	1.8 (±0.98)
Average number of antibiotics used among patients since admission			
Mean (±SD)	1.3 (±1.56)	1.2 (±0.46)	1.2 (±0.53)
**Indications for antimicrobial used** (*n* = 1099)			
Medical prophylaxis	317 (44.3)	235 (53.3)	552 (47.7)
Community-acquired infection	234 (32.7)	134 (30.4)	368 (31.8)
Hospital-acquired infection	5 (0.7)	4 (0.9)	9 (0.8)
Surgical prophylaxis	191 (26.7)	73 (16.6)	264 (22.8)
**Dose and duration of antibiotics used as surgical prophylaxis (SP)**			
One dose on 1 day/multiple days (SP1)	5 (2.6)	2 (2.7)	7 (2.7)
Multiple doses on 1 day (SP2)	3 (1.6)	9 (12.3)	12 (4.6)
Multiple doses > 1 day (SP3)	183 (95.8)	62 (84.9)	245 (92.8)

SD—Standard deviation; SP—Surgical Prophylaxis.

**Table 2 antibiotics-11-00810-t002:** Use of antibiotics according to demographic and clinical characteristics of enrolled patients at tertiary and secondary level hospitals in Bangladesh from February to April 2021.

Characteristics	Antibiotic Use in Tertiary Level Hospitals (N = 892) *n* (%)	Antibiotic Use in Secondary Level Hospitals (N = 525) *n* (%)	*p*-Value	Overall Antibiotic Use (N = 1417) *n* (%)
**Sex of patients**				
Female	338 (74.3)	227 (82.3)	0.013	565 (77.3)
Male	331 (75.7)	203 (81.5)	0.080	534 (77.8)
**Age group**				
0–14 years	214 (85.3)	145 (98.0)	0.000	359 (90.0)
15–64 years	407 (72.6)	248 (76.1)	0.249	655 (73.8)
65 years and above	48 (60.0)	37 (72.6)	0.142	85 (64.9)
**Departments**				
Medicine	168 (62.5)	145 (71.1)	0.050	313 (66.2)
Surgery	189 (75.0)	123 (83.1)	0.059	312 (78.0)
Gynae and Obstetrics	118 (78.7)	66 (86.8)	0.136	184 (81.4)
Pediatrics	194 (87.8)	96 (99.0)	0.001	290 (91.2)
**Had surgery after admission ***	161 (97.6)	57 (100.0)	0.236	218 (98.2)
**Use of Devices**				
Peripheral vascular catheter	545 (84.0)	340 (85.4)	0.528	885 (84.5)
Urinary catheter	90 (87.4)	23 (85.2)	0.763	113 (86.9)
Central vascular catheter	7 (63.6)	4 (100.0)	0.159	11 (73.3)
Intubation device	4 (100.0)	0 (0.0)	0.025	4 (80.0)
**History of transfer or hospitalization**				
Patients transferred from another hospital	180 (82.6)	35 (89.7)	0.264	215 (83.7)
Hospitalization within the last 90 days	97 (70.3)	47 (74.6)	0.529	144 (71.6)
**Indication for antimicrobial use**				
Medical Prophylaxis	293 (92.4)	223 (94.9)	0.246	516 (93.5)
Community-Acquired Infections	213 (91.0)	130 (97.0)	0.028	343 (93.2)
Surgical Prophylaxis	189 (99.0)	73 (100.0)	0.380	262 (99.2)
Hospital-Acquired Infections	5 (100.0)	4 (100.0)		9 (100.0)

* Denominators are taken from Table 1.

**Table 3 antibiotics-11-00810-t003:** Distribution of antimicrobial use according to different departments and types of hospitals in Bangladesh from February to April 2021.

Antimicrobials Used	WHO AWaRe Classification	Number of Antimicrobial Agents Used According to Different Departments				Number of Antimicrobial Agents Used According to the Types of Hospitals		Overall Antimicrobial Agents Used (N = 2138)
		Medicine (N = 463)	Surgery (N = 611)	Gynae and Obstetrics (N = 479)	Pediatrics (N = 585)	Tertiary Level Hospitals (N = 1416)	Secondary Level Hospitals (N = 722)	
		*n* (%)	*n* (%)	*n* (%)	*n* (%)	*n* (%)	*n* (%)	*n* (%)
Antibiotics (In total)		455 (98.3)	607 (99.3)	478 (99.8)	572 (97.8)	1397 (98.7)	715 (99.0)	2112 (98.8)
Cephalosporins Group (In total)		241 (52.1)	347 (56.8)	213 (44.5)	285 (48.7)	693 (48.9)	393 (54.4)	1086 (51.4)
1st-generation cephalosporins	Access	0 (0.0)	0 (0.0)	23 (4.8)	0 (0.0)	23 (1.6)	0 (0.0)	23 (1.1)
2nd-geneneration cephalosporins	Watch	22 (4.8)	37 (6.1)	31 (6.5)	3 (0.5)	45 (3.2)	48 (6.6)	93 (4.3)
3rd-geneneration cephalosporins	Watch	217 (46.9)	304 (49.8)	158 (33.0)	275 (47.0)	622 (43.9)	332 (46.0)	954 (44.6)
4th-geneneration cephalosporins	Watch	2 (0.4)	4 (0.7)	1 (0.2)	7 (1.2)	2 (0.1)	12 (1.7)	14 (0.7)
2nd-geneneration cephalosporins + beta lactamase inhibitors	Not recommended	0 (0.0)	2 (0.3)	0 (0.0)	0 (0.0)	1 (0.1)	1 (0.1)	2 (0.1)
Imidazoles	Access	34 (7.3)	86 (14.1)	121 (25.3)	12 (2.1)	172 (12.1)	81 (11.2)	253 (11.8)
Penicillins	Access	11 (2.4)	116 (19.0)	56 (11.7)	80 (13.7)	167 (11.8)	96 (13.3)	263 (12.3)
Aminoglycosides	Access	2 (0.4)	13 (2.1)	7 (1.5)	132 (22.6)	137 (9.7)	17 (2.4)	154 (7.2)
Macrolides	Watch	82 (17.7)	3 (0.5)	31 (6.5)	8 (1.4)	60 (4.2)	64 (8.9)	124 (5.8)
Fluoroquinolones	Watch	25 (5.4)	23 (3.8)	40 (8.4)	8 (1.4)	53 (3.7)	43 (6.0)	96 (4.5)
Carbapenems	Watch	9 (1.9)	10 (1.6)	4 (0.8)	41 (7.0)	52 (3.7)	12 (1.7)	64 (3.0)
Beta lactam-beta lactamase inhibitors	Access	41 (8.9)	2 (0.3)	1 (0.2)	1 (0.2)	42 (3.0)	3 (0.4)	45 (2.1)
Glycopeptides	Watch	0 (0.0)	0 (0.0)	0 (0.0)	5 (0.9)	5 (0.4)	0 (0.0)	5 (0.2)
Lincosamide	Access	2 (0.4)	3 (0.5)	4 (0.8)	0 (0.0)	5 (0.4)	4 (0.6)	9 (0.4)
Furadantin	Access	3 (0.6)	1 (0.2)	0 (0.0)	0 (0.0)	3 (0.2)	1 (0.1)	4 (0.2)
Anti-tubercular agents	Not- classified	3 (0.6)	1 (0.2)	0 (0.0)	0 (0.0)	4 (0.3)	0 (0.0)	4 (0.2)
Rifamycins	Watch	1 (0.2)	1 (0.2)	0 (0.0)	0 (0.0)	2 (0.1)	0 (0.0)	2 (0.1)
Oxazolidinones	Reserve	0 (0.0)	1 (0.2)	1 (0.2)	0 (0.0)	1 (0.1)	1 (0.1)	2 (0.1)
Trimethoprim-sulfonamide combinations	Access	1 (0.2)	0 (0.0)	0 (0.0)	0 (0.0)	1 (0.1)	0 (0.0)	1 (0.0)
Antivirals	Non-Antibiotics	6 (1.3)	2 (0.3)	1 (0.2)	10 (1.7)	15 (1.1)	4 (0.6)	19 (0.9)
Antifungals	Non-Antibiotics	0 (0.0)	1 (0.2)	0 (0.0)	1 (0.2)	2 (0.1)	0 (0.0)	2 (0.1)
Antiparasitics	Non-Antibiotics	2 (0.4)	0 (0.2)	0 (0.0)	2 (0.3)	1 (0.1)	3 (0.4)	5 (0.2)
Total		463 (100.0)	611 (100.0)	479 (100.0)	585 (100.0)	1416 (100.0)	722 (100.0)	2138 (100.0)

## Data Availability

The data presented in this survey are available on reasonable request from icddr’b’s research administration through the corresponding author. The data are not publicly available due to privacy restrictions and icddr,b policy.

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
