# Peer review of "Pattern of Antibiotic Use among Hospitalized Patients according to WHO Access, Watch, Reserve (AWaRe) Classification: Findings from a Point Prevalence Survey in Bangladesh"

_antibiotics, 2022, doi:10.3390/antibiotics11060810_

Round 1

Reviewer 1 Report

The manuscript entitled " Pattern of antibiotic use among hospitalized patients according to WHO Access, Watch, Reserve (AWaRe) Classification: Findings from a Point Prevalence Survey in Bangladesh " is an interesting and well conducted multicenter cross-sectional survey about in-hospital antimicrobial use in Bangladesh. Overall, the importance of the explored issue and the originality of the data deserve publication.

However, the manuscript requires English editing, particularly in the introduction and methods section.

I would suggest to:

- Define, if possible, the reasons why medical prophylaxis was prescribed

- Provide, if possible, the mean duration of antibiotic therapy per prescriptive category (SP, MP, CAI, HAI)

- Reframe Table 3 and Figure 3

- Uniform the use of bold type in Table 1

I would also suggest implementing the reference section with the followings:

- Ioannou P, Karakonstantis S, Schouten J, Kostyanev T, Charani E, Vlahovic-Palcevski V, Kofteridis DP; supported by the ESCMID Study Group for Antimicrobial Stewardship (ESGAP). Indications for medical antibiotic prophylaxis and potential targets for antimicrobial stewardship intervention: a narrative review. Clin Microbiol Infect. 2022 Mar;28(3):362-370. doi: 10.1016/j.cmi.2021.10.001. Epub 2021 Oct 12. PMID: 34653572.

- Antimicrobial Resistance Collaborators. Global burden of bacterial antimicrobial resistance in 2019: a systematic analysis. Lancet. 2022 Feb 12;399(10325):629-655. doi: 10.1016/S0140-6736(21)02724-0. Epub 2022 Jan 19. PMID: 35065702; PMCID: PMC8841637.

- Segala FV, Murri R, Taddei E, Giovannenze F, Del Vecchio P, Birocchi E, Taccari F, Cauda R, Fantoni M. Antibiotic appropriateness and adherence to local guidelines in perioperative prophylaxis: results from an antimicrobial stewardship intervention. Antimicrob Resist Infect Control. 2020 Oct 26;9(1):164. doi: 10.1186/s13756-020-00814-6. PMID: 33106190; PMCID: PMC7586646.ù

Author Response

Point 1: Define, if possible, the reasons why medical prophylaxis was prescribed

Response 1: The definition is given in the manuscript in the line numbers 122 to 124.

Point 2:  Provide, if possible, the mean duration of antibiotic therapy per prescriptive category (SP, MP, CAI, HAI)

Response: We did not collect the duration of antibiotic therapy prescribed by the physicians. 

Point 3: Reframe Table 3 and Figure 3

Response: Reframing is done for Table 3 and Figure 2

Point 4: Uniform the use of bold type in Table 1

Response: Uniformity is done in Table 1

Point 5:  I would also suggest implementing the reference section with the followings:

- Ioannou P, Karakonstantis S, Schouten J, Kostyanev T, Charani E, Vlahovic-Palcevski V, Kofteridis DP; supported by the ESCMID Study Group for Antimicrobial Stewardship (ESGAP). Indications for medical antibiotic prophylaxis and potential targets for antimicrobial stewardship intervention: a narrative review. Clin Microbiol Infect. 2022 Mar;28(3):362-370. doi: 10.1016/j.cmi.2021.10.001. Epub 2021 Oct 12. PMID: 34653572.

- Antimicrobial Resistance Collaborators. Global burden of bacterial antimicrobial resistance in 2019: a systematic analysis. Lancet. 2022 Feb 12;399(10325):629-655. doi: 10.1016/S0140-6736(21)02724-0. Epub 2022 Jan 19. PMID: 35065702; PMCID: PMC8841637.

- Segala FV, Murri R, Taddei E, Giovannenze F, Del Vecchio P, Birocchi E, Taccari F, Cauda R, Fantoni M. Antibiotic appropriateness and adherence to local guidelines in perioperative prophylaxis: results from an antimicrobial stewardship intervention. Antimicrob Resist Infect Control. 2020 Oct 26;9(1):164. doi: 10.1186/s13756-020-00814-6. PMID: 33106190; PMCID: PMC7586646.ù

Response 5: Added in the reference and in the writings.

Reviewer 2 Report

This is an interesting and well-conducted study. I believe figures, tables and references are appropriate. Only have minor suggestions for the authors:

1. add p-value in abstract line 33 as this is important finding

2. line 36 define ABU

3. line 130 more data on STATA should be presented

4. line 142 what is icddr,b

5. please add abreviation under the tables

Author Response

Point 1: add p-value in abstract line 33 as this is important finding

Response 1: Added in the line number 36

Point 2: line 36 define ABU

Response 2: The definition of ABU is given in the line number 38.

Point 3: line 130 more data on STATA should be presented

Response 3: Added in the line number 135

Point 4: line 142 what is icddr,b

Response 4: Elaborated in the lines numbers 147 and 148 

Point 5: please add abbreviations under the tables

Response 5: Added  under table 1